# Characterization of a PIP Binding Site in the N-Terminal Domain of V-ATPase a4 and Its Role in Plasma Membrane Association

**DOI:** 10.3390/ijms24054867

**Published:** 2023-03-02

**Authors:** Anh Chu, Yeqi Yao, Golam T. Saffi, Ji Hyun Chung, Roberto J. Botelho, Miroslawa Glibowicka, Charles M. Deber, Morris F. Manolson

**Affiliations:** 1Faculty of Dentistry, University of Toronto, Toronto, ON M5G 1G6, Canada; 2Department of Pharmacology and Toxicology, University of Toronto, Toronto, ON M5S 1A8, Canada; 3Faculty of Arts and Science, Queen University, Kingston, ON K7L 3N9, Canada; 4Department of Chemistry and Biology, Toronto Metropolitan University, Toronto, ON M5G 2A7, Canada; 5Division of Molecular Medicine, Research Institute, Hospital for Sick Children, Toronto, ON M5G 1X8, Canada; 6Department of Biochemistry, Faculty of Medicine, University of Toronto, Toronto, ON M1C 1A4, Canada

**Keywords:** V-ATPases, V-ATPase a4 isoforms, protein–lipid interaction, phosphoinositides, PI(4,5)P_2_, dRTA

## Abstract

Vacuolar ATPases (V-ATPases) are multi-subunit ATP-dependent proton pumps necessary for cellular functions, including pH regulation and membrane fusion. The evidence suggests that the V-ATPase a-subunit’s interaction with the membrane signaling lipid phosphatidylinositol (PIPs) regulates the recruitment of V-ATPase complexes to specific membranes. We generated a homology model of the N-terminal domain of the human a4 isoform (a4NT) using Phyre2.0 and propose a lipid binding domain within the distal lobe of the a4NT. We identified a basic motif, K^234^IKK^237^, critical for interaction with phosphoinositides (PIP), and found similar basic residue motifs in all four mammalian and both yeast a-isoforms. We tested PIP binding of wildtype and mutant *a4*NT in vitro. In protein lipid overlay assays, the double mutation K234A/K237A and the autosomal recessive distal renal tubular-causing mutation K237del reduced both PIP binding and association with liposomes enriched with PI(4,5)P_2_, a PIP enriched within plasma membranes. Circular dichroism spectra of the mutant protein were comparable to wildtype, indicating that mutations affected lipid binding, not protein structure. When expressed in HEK293, wildtype a4NT localized to the plasma membrane in fluorescence microscopy and co-purified with the microsomal membrane fraction in cellular fractionation experiments. a4NT mutants showed reduced membrane association and decreased plasma membrane localization. Depletion of PI(4,5)P_2_ by ionomycin caused reduced membrane association of the WT a4NT protein. Our data suggest that information contained within the soluble a4NT is sufficient for membrane association and that PI(4,5)P_2_ binding capacity is involved in a4 V-ATPase plasma membrane retention.

## 1. Introduction

Vacuolar ATPases (V-ATPases) are ATP-dependent proton pumps that maintain the luminal pH of intracellular organelles [1,2,3,4]. V-ATPases in specialized cells, such as epididymis, kidney and bone cells, are targeted to the plasma membrane for extracellular pH regulation [5,6,7,8], or in the case of metastasizing cells, for extracellular acidification that enables invasiveness [9,10,11]. Eukaryotic V-ATPases consist of 14 subunits distributed into two sectors (V_1_ and V_o_), with many subunits having multiple isoforms [1,2,12,13]. V_1_ contains nucleotide binding sites and is responsible for ATP hydrolysis. ATP hydrolysis creates a driving force, resulting in the rotation of a membrane-bound barrel, composed of the *c* subunits in the V_o_ sector, in turn leading to the transport of protons across the membrane, from the cytoplasm to the luminal space [12].

Each V-ATPase complex contains one copy of the ~100 kDa a subunit which exists in mammalian cells as 4 isoforms (a1, a2, a3 and a4). Human mutations in the genes encoding the a1, a2, a3 and a4 have been associated with epileptic encephalopathy [14], cutis laxa [15,16], osteopetrosis [6,17,18,19] and renal tubular acidosis [5,20], respectively. The ~50 kDa N-terminus of a-subunit is oriented parallel to the membrane and is in contact with V_1_ subunits, critical for coupling V_1_ ATP hydrolysis with V_o_ proton translocation [13]. Studies with the yeast homologs of a-subunit, Vph1p and Stv1p, suggest targeting information is in the cytosolic N-terminal domain of this subunit [21]. Recent evidence indicates a direct interaction between Vph1p/Stv1p and different membrane phosphoinositides (PIPs) to ensure the localization and/or activation of yeast V-ATPases. The N-terminal domain of Vph1p has been shown to interact with vacuolar membrane PI(3,5)P_2_, and in the absence of PI(3,5)P_2_ Vph1p fails to localize to vacuoles [22,23]. Whether human subunit *a* isoforms interact with PIPs and if such interactions affect differential subcellular membrane distributions and/or regulation of mammalian V-ATPases is not known.

PIPs are phosphorylated derivatives of inositol phospholipids (PI) that are concentrated at the cytosolic leaflet of the membrane [24]. Reversible phosphorylation of hydroxyl groups at different positions on the inositol ring result in seven PIP species. Different kinases and phosphatases involved in PIPs biosynthesis have discreet subcellular localization, resulting in a characteristic enrichment of PIPs on specific organelles [25,26]. Organelle-specific distribution of PIPs help define organelle identity, crucial in membrane protein trafficking [24,27]. PI(4,5)P_2_ is enriched in the inner leaflet of the plasma membrane, and participates in multiple cell surface events [28,29,30]. PI(4,5)P_2,_ is a substrate for phosphoinositide phospholipase C (PLC), which generates the second messenger Ins(3,4,5)P_3_ and diacylglycerol (DAG), metabolites required for the propagation of multiple signaling pathways [28,30,31,32,33,34]. Dephosphorylation of PI(4,5)P_2_ by synaptojanin 1, a PI 5-phosphatase, is crucial for cancer cell invasion and migration [35,36]. PI(4,5)P_2_ is recognized as a “signpost” to sequester proteins involved in the modulation of actin polymerization and endocytosis to the plasma membrane [37,38,39].

Different subunit a-isoforms are responsible for targeting V-ATPases to various subcellular membranes [21,40]. a3 and a4 are enriched in the plasma membrane of osteoclasts [19] and renal intercalated cells [5], respectively. Further, upregulation of both a3 and a4 have been linked to the invasiveness of breast cancer cells [9,11]. a4 was localized to the membrane of invadopodia of mouse breast cancer cells, where it is involved in invasion and migration [41]. Mutations in the a4-encoding gene *Atp6v0a4* cause autosomal recessive distal renal tubular acidosis (dRTA), characterized by impaired urine acidification leading to severe hyperchloremic metabolic acidosis and impaired bone physiology and hearing [20,42,43,44].

Here, we show that a4NT interacts with PI(4,5)P_2_ in vitro and that disruption of this lipid binding results in reduced plasma membrane localization of the protein in HEK293 cells. Using molecular modeling, sequence alignment and mutagenesis, we propose a putative PIP binding site with the conserved basic motif, K^234^IKK^237^, at the distal lobe of the a4 N-terminal domain. The K237del mutation in this motif was identified in a female patient with dRTA [45]. We created two mutations, the human mutation K237del and K234A/K237A, and found that both mutations reduced PI(4,5)P_2_ binding and a4NT plasma membrane localization.

## 2. Results

### 2.1. Homology Model of a4NT Revealed a Putative Lipid Binding Motif K^234^IKK^237^ Conserved in Yeast and All Four Mammalian Isoforms

We used Phyre2.0 to generate a three-dimensional structure model of the cytoplasmic N-terminal domain of the human a4 isoform (a4NT) with the available cryo-electron-microscopy-generated structure of yeast Vph1p-NT (PDB: 3j9t.b) (Figure 1A), which shares 32% identity with the a4NT isoform. Using the structure alignment, we identified a putative PIP binding site in the distal lobe of a4NT (Figure 1A, inset), which shares a similar secondary structure arrangement with the basic pocket of the PH domain in Osh3, a defined PI(4)P-binding protein (PDB: 4ic4). Compared to the Osh3 PH domain [46], the putative binding site in a4 adopts a more compact incomplete barrel structure, consisting of three parallel β-strands and two α-helices. Considering the importance of basic or aromatic amino acids in PIP interaction in non-canonical PIP binding domains, we identified the basic motif K^234^IKK^237^ within the putative binding domain (Figure 1B). This basic motif, which may be critical for the interaction with the PIP acidic head groups, is accessible to the plasma membrane lipid and is conserved in all four mammalian NTs, as well as yeast Vph1p and Stv1p. There are basic/aromatic residues variation within this motif which we hypothesize confers PIP specificity to different *a* isoforms (Figure 2A).

### 2.2. Mutation within the Putative a4-PIP Binding Motif Reduced In Vitro PIP Binding and Interactions with PI(4,5)P_2_-Enriched Liposomes

To assess our putative PIP binding motif, we generated two mutations in a4NT: K234A/K237A and K237del. Both mutations are in the putative binding motif and K237del is a human mutation causing distal renal acidosis [44].

We tested in vitro PIP interactions of wildtype and mutant a4NTs with a protein-lipid overlay assay and a PIP-enriched liposome pull-down assay. The PIP overlay assay demonstrated significant decreases in the PIP binding of both mutants compared to wildtype (Figure 2B,C). The results do not show any statistically significant PIP specificity for the a4 isoform. This may reflect the dimensional constraints of the PIPs being blotted onto the nitrocellulose membrane in the overlay assay; membrane curvature is often required for lipid specificity in PIP interaction [47,48] and is absent in overlay blot assays.

As a4-containing V-ATPases are targeted to the plasma membrane of renal intercalated cells, we were particularly interested in the interaction of the a4 isoform with plasma membrane PIPs. To mimic the plasma membrane environment and structure, we used PolyPIPosomes liposomes enriched with PI(3,4)P_2_, PI(4,5)P_2_ and PIP3, which are mostly found in the plasma membrane. Using liposome pull-down assays, wildtype and mutant K234A/K237A were incubated with these different PolyPIPosomes. While the K234A/K237A mutants reduced association with all three liposomes, only the associations with the PI(4,5)P_2_-enriched liposomes were statistically significant (Figure 3A,B). PI(4,5)P_2_ is concentrated in the plasma membrane and plays a major role in both the regulation and recruitment of plasma membrane proteins. We also assessed the binding of the human mutation K237del with PI(4,5)P_2_-enriched liposomes; we observed a significant decrease in the liposome binding of the deletion mutant compared to wildtype (Figure 3B). Interestingly, when testing the binding of a4 proteins to PI(4)P, a PIP not enriched in the plasma membrane but rather enriched in the Golgi, there was no significant difference in the binding between wildtype and K237del (Figure 3C).

To ask whether the mutations affected PIP binding or overall protein structure, we assessed protein folding using circular dichroism (CD) spectroscopy of both WT and mutant protein in an aqueous environment, in the presence and absence of SDS (Figure 2D). WT a4NT contains largely a helical structure, exhibited by two negative minima at 222 nm and 208 nm, consistent with the structural model of a4NT. The CD spectra of the mutants were comparable to the wildtype with overlapping peaks at those specific wavelengths, indicating that the mutations disrupt PIP binding without altering protein structure (Figure 2D). In CD spectroscopy, the presence of micelles enhance helicity characteristics of a membrane-bound protein by creating a membrane-like environment. In this study, we added 10 mM SDS (SDS to peptide ratio was 370:1) to determine whether the presence of SDS micelles could increase the helicity of the proteins. The CD spectra of both WT and mutant a4 showed deeper minima at 208 nm and 222 nm, suggesting that the cytosolic WT and mutant proteins have membrane-bound protein characteristics (Figure 2D). The non-overlapping positive spectra at 190–200 nm could be accounted for by differences in the trace amount of salt or imidazole in the protein buffer. As the secondary protein structure appears unaffected by the mutations, our results suggest that the K^234^IKK^237^ motif is, in part, involved in the interaction with PIPs, and mutations within this motif reduce interaction with PI(4,5)P2.

### 2.3. K234A/K237A and K237del Mutations Reduce a4NT-Membrane Association In Vivo

To assess the effect of a4NT mutations on membrane association in vivo, the FLAG-tagged wildtype and mutants K234A/K237A and K237del a4NT were expressed in HEK293 cells and subcellular fractionation experiments were performed to determine whether the mutations affected protein retention in microsomal fractions. There was a statistically significant decrease in the amount of K237del and K234A/K237A mutants in the microsomal fraction compared to wildtype (Figure 4A,B).

Immunofluorescence microscopy was used to assess whether the mutations affect membrane localization of the proteins. Wildtype FLAG-tagged a4NT (green) are enriched in the vicinity of the plasma membrane, labelled with wheat germ agglutinin (WGA, red) (Figure 5A, top panel). This indicates that even though a4NT lacks a transmembrane domain, cytosolic a4NT contains sufficient information for plasma membrane retention. By contrast, both mutant proteins were mainly in the cytosolic fraction and did not appear in the vicinity of the plasma membrane (Figure 5A, middle horizontal panels). Quantification by assessing protein signal intensity at the vicinity of the plasma membrane marker revealed a 75% decrease in signal detected at the plasma membrane for both mutants (Figure 5B), consistent with the in vitro data above.

### 2.4. Ionomycin Reduces a4NT-Membrane Association In Vivo

We have observed that mutations within the putative lipid binding motif disrupt membrane association both in vitro and in vivo. As further proof of PIP involvement, we then asked whether modification of the plasma membrane phosphoinositides could affect a4NT membrane localization. Treatment of cells with ionomycin, a Ca^2+^ ionophore, leads to the activation of PLCs, resulting in the rapid breakdown of PI(4,5)P_2_ [49]. To analyze the correlation between PI(4,5)P2 changes and the redistribution of the a4NT at the plasma membrane, HEK293 cells transfected with FLAG-tagged wildtype a4NT were treated with 5 µM of ionomycin at room temperature for 15 min. Wildtype a4NT (red) appear in some regions of the plasma membrane, labelled with anti-Na^+^/K^+^-ATPase antibodies (green); ionomycin treatment reduced wildtype a4NT signal at the plasma membrane (Figure 6A). Quantification was performed by dividing the fluorescence of the membrane-associated WT signal with total protein fluorescence. There was an approximately 50% reduction in a4NT associated with membrane in cells treated with ionomycin (Figure 6B) suggesting that depletion of the plasma membrane PI(4,5)P_2_ reduces a4NT membrane association. As a control, we showed that plasma membrane localization of the PH domain of PLC, an established PI(4,5)P_2_-binding protein, was similarly reduced in the presence of ionomycin (Figure 6C).

## 3. Discussion

Here, we show that the V-ATPase a4 isoform, predominantly localized in the plasma membrane of the renal intercalated cells in vivo, interacts with PI(4,5)P_2_, a PIP concentrated on the plasma membrane. To support the physiological significance of this interaction, mutations within a putative PIP binding site reduced the a4-PI(4,5)P_2_ interaction. Further, reduction of PI(4,5)P_2_ levels decreased a4NT plasma membrane localization. Studies in yeast have shown that V-ATPases containing the vacuolar isoform Vph1p can be recruited to membranes in a PI(3,5)P_2_-dependent manner [23]. These data all suggest that there is a PIP interaction site within the cytosolic N-terminal domain of the a-subunits that recognizes phosphoinositides in specific membranes and plays a role in the spatial regulation of V-ATPases.

Pleckstrin homology (PH) domains are a major type of membrane binding domains, and have been well characterized as binding modules to PIPs [50]. They consist of two β-sheets curving to form a barrel with basic key residues within the flexible loops connecting the β-sheet, and the barrel structure is enclosed by a C-terminal α-helix. However, many membrane proteins bind to PIPs without a canonical domain but rather use positively charged patches. Here, we proposed a putative PIP binding domain in the distal lobe of the N-terminal domain of the a-subunit, which has a similar structural arrangement to the basic pocket for PI headgroup binding in the Osh3 protein [46]. Screening for basic or aromatic patches that facilitate the interaction with PIP headgroup, we identified the basic motif, (K/R)X(K/R)(K/R), within the putative binding domain. This motif is conserved in all four human a-subunit isoforms and yeast Vph1p and Stv1p, with some variations in the amino acid composition, which, we hypothesize, confers the specificity of different isoforms (Figure 2D). Interestingly, fitting of the a4NT homology model in the completed structure of V-ATPases showed that the side chain of the basic residues oriented outward, allowing access to membrane PIP headgroups (Figure 1B). Furthermore, mutations within these motifs have been identified in V-ATPase-related diseases, including a4.K237del found in patients with distal renal tubular acidosis [45], and a2. K237_V238del associated with cutis laxa [15]. Taken together, these data indicate a basic motif involved in V-ATPase a-subunit-PIP interactions and V-ATPases regulation. In a yeast study, Banerjee et al. suggested that residue K84 within Stv1p is critical for PI(4)P interaction [22]. The K84 residue resides on a flexible loop exposed to the membrane and is within the Stv1p Golgi localization signal—W^83^KY [21]; however, this residue is not conserved in humans and Vph1p. Studies of PIP binding domains indicate that highly conserved basic motifs are necessary, but not exclusive, for interaction with PIPs [51]. Further, protein folding plays an additional key role in bringing distant residues in the sequence to closer proximity in 3D structure, which altogether strengthens protein–lipid interactions. We hypothesize that in Stv1p, the putative binding motif and the K84 residues work cooperatively to strengthen and/or define the specificity of the protein and lipid interaction.

We present evidence that the cytosolic N-terminal domain of the a4 is sufficient for membrane association and that mutations within the motif K^234^IKK^237^ disrupt binding to the plasma membrane enriched PIP, PI(4,5)P_2_. This is consistent with previous work which showed that the targeting signal for V-ATPases lies within the N-terminal domain of the a-subunit [21] and V-ATPases containing different isoforms of the a-subunit are targeted at different locations within the cells. We hypothesize that the variation in the basic motifs may, in part, account for the different PIP specificity of the four isoforms, conferring differential regulation of V-ATPases at specific membranes.

Each of the seven PIP species has a unique cellular distribution that renders them a lipid address for different organelles. Multiple studies have shown that PIPs are key regulators in membrane trafficking, responsible for recruitment of signaling effectors. Here, we show that reducing plasma membrane PI(4,5)P_2_ concentrations using ionomycin decreased a4NT plasma membrane localization of the *a* subunit. Studies using the metastatic breast cancer cell lines MBA-231 showed that enhanced a3 and a4 expression and upregulation of PI(4,5)P_2_ were both critical for formation of invadopodia and cell invasion [52]; this infers that PIP–a-subunit interactions are required for V-ATPase involvement in metastasizing cells and suggests the PIP binding site as a novel target for cancer therapeutics. In summary, we propose a lipid binding domain within the N-terminal domain of the V-ATPase a-subunit and present in vitro and in vivo evidence that a4NT can bind to PIPs through this domain. Further studies are now required to elucidate whether PIP-V-ATPase interactions confer either spatial or enzymatic regulation or a combination of both.

## 4. Materials and Methods

### 4.1. Plasmids

**pET32a+::human V-ATPase a4NT (MM1111)**. The PCR products of N-terminal domain of human a4 with primers MO503: 5′-ACGTGGTACCATGGTGTCTGTGTTTCGAAG and MO504: 3′-ACGTGAATTCCTCCGGTGTAGGGGGCTGGGTTTATC, from start codon to T397, was cloned into pcDNA3 and pET32a(+) plasmids between *KpnI* and *EcoRI* sites, respectively, resulting in MM1071 and MM1103; and PCR products of human a4 with primers MO521: 5′-GCAACATCGACGTCACCCAGCAG-3′ and MO522: 5′-ACGTGAATTCTTAGGTGTAGGGGGCTGGGTTTATC-3′ was used to replace the region from *ZraI* to *EcoRI* (at position V316 to the 3′ end) in MM1103, which adding a stop codon TAA just after T397, resulting in MM1111. **pET32a+::human V-ATPase a4NT K234A/K237A (MM1113)**. A designed a4NT K234A/K237A double mutation was synthesized by GeneArt Gene Synthesis, Invitrogen Canada, and delivered in the form of pMA-T::partial a4NT DNA fragment covered from *BamHI* to *EcoNI* sites of human a4NT. The synthesized DNA fragment was used to replace the same region of MM1071, resulting in MM1097; when the whole *KpnI* and *EcoRI* insert of a4NT in MM1097 was moved into pET32a(+), the new construct was named as MM1105. A TAA stop codon after T397 was added in MM1105 and resulted in the final MM1113. **pET32a+::human V-ATPase a4NT ∆K237 (MM1130)**. A Q5 Site-Directed Mutagenesis Kit (NEB E0554) was used on MM1111 to make human V-ATPase a4NT.K237del mutant with primers MO527: 5′-ATCTGTGATGGGTTTCGAG and MO528: 3′-CTTGATTTTCTGCCTGAG, sequencing verified. **pcDNA3::human V-ATPase a4NT (MM1123)**. *KpnI* and *EcoRI* insert of a4NT in MM1111 was moved into pcDNA3.1+ and the new construct was named MM1123. **pcDNA3::human V-ATPase a4NT K234A/K237A (MM1125)**. *KpnI* and *EcoRI* insert of MM1113 was ligated into pcDNA3.1+ and the new construct was named MM1125. **pcDNA3::human V-ATPase a4NT ∆K237 (MM1131)**. The *KpnI* and *EcoRI* insert of MM1130 was ligated into pcDNA3.1+ and the new construct was named MM1131.

### 4.2. Expression and Purification of Human a4NT Wildtype and Mutants from E.coli

N-terminal domain of human a4 (amino acid 1-421) with 6His tag was expressed in *E.coli* Rosetta (DE3) (Sigma-Aldrich, St. Louis, MO, USA) via pET32a plasmid. Cells were grown in LB media with ampicillin (100 μg/mL) and chloramphenicol (34 μg/mL) at 37 °C until OD600 = 0.7~0.8. Expression was induced with 0.2 mM isopropyl 1-thio-b-D-galactopyranoside (IPTG) (Thermo Fisher, Waltham, MA, USA), cultured at 16 °C in a shaker bath overnight. Cells were harvested by centrifugation, resuspended in lysis buffer (containing 50 mM NaH_2_PO_4_ pH 7.8, 300 mM NaCl, EDTA-free protease inhibitor tablet (Roche, Basel, Switzerland), 2 mM PMSF, 3 mM 2-mercaptoethanol, 5 μg/mL DNase, 5 μg/mL RNase, 1% FC-12 (Anatrace, Maumee, OH, USA)), and lysed by shaking in a cold room for 1 h. Lysate were cleared by centrifugation (40,000 rpm) and passed over a Ni-NTA Agarose (Qiagen, Hilden, Germany) column. The column was washed with 50 mM NaH_2_PO_4_ pH 7.8, 300 mM NaCl, 20 mM imidazole and 0.01% FC-12. Protein was eluted with 10 resin volumes of elution buffer (50 mM NaH_2_PO_4_ pH 7.8, 300 mM NaCl, 200 mM imidazole), and dialyzed against 50 mM Tris pH 8.0, 150 mM NaCl and 1 mM DTT overnight. The protein was further concentrated by ultrafiltration using an Amicon Ultra-50 Centrifugal Filter Unit.

### 4.3. Protein Structure Prediction

A homology model of the N-terminus of the a4 isoform (amino acid 1-420) was constructed using Phyre2.0 with reference coordinates of *S. cerevisiae* V-ATPase state 1 Vph1p-NT (PDB: 3j9t.b). Confidence interval of the model was 100%. Sequence alignment of all 4 human a-isoforms and yeast Vph1p, Stv1p were obtained by Chimera.

### 4.4. Protein–Lipid Overlay Assay

PIP Array membranes (Echelon, Santa Clara, CA, USA) were blocked overnight at 4 °C in blocking buffer (10 mM Tris pH 8.0, 150 mM NaCl, 0.1% Tween-20, 3% fatty acid-free BSA). A total of 12 μg of purified proteins was diluted in 10 mL blocking buffer and incubated with the membrane for 2 h at 4 °C. The membranes were washed 3 times for 10 min with cold TBS-T (10 mM Tris pH 8.0, 150 mM NaCl, 0.1% (*v/v*) Tween-20). PIP membranes were then incubated with mouse anti-His antibody (1:2000 dilution) in blocking buffer for 1 h, washed 3 times in cold TBS-T, and incubated with horseradish peroxidase-conjugated anti-mouse immunoglobulin G in blocking buffer for 1h. After 3 washes with cold TBS-T, membranes were developed by enhanced chemiluminescence immunoblotting detection reagent (GE Healthcare, Chicago, IL, USA) and exposed for 30 s.

### 4.5. PolyPIPosome Pull-Down Assay

A total of 10 μg of purified proteins was incubated with 20 µL 1 mM PI(4,5)P_2_-PolyPIPosome (Echelon) and 200 µL of binding buffer (50 mM Tris pH 8.0, 150 mM NaCl and 0.05% Nonidet P-40) and incubate rotating at 4 °C for 2 h. The protein-bound liposomes were pelleted by centrifugation at 13,000 rpm for 10 min, followed by 3 washes with 200 µL of binding buffer each. Pellets were resuspended in 20 μL of binding buffer and 20 μL of 2× SDS sample buffer, separated by SDS-PAGE and transferred to nitrocellulose membrane. Membranes were then incubated with mouse anti-His antibody (1:5000 dilution) in TBS-T for 1 h, washed another 3 times with TBS-T and incubated with horseradish-peroxidase-conjugated anti-mouse immunoglobulin G in TBS-T for 1 h. After 3 washes with TBS-T, membranes were developed by enhanced chemiluminescence immunoblotting detection reagent (GE Healthcare) and exposed for 5–10 s.

### 4.6. HEK293 Transfection and Cellular Fractionation

HEK293 cells (ATCC) were cultured on 10 cm culture dishes in Dulbecco’s modified Eagle’s medium (DMEM) (Gibco) containing 10% fetal bovine serum (FBS) and 0.5% antibiotics, and grown in a 95% air, 5% CO_2_ humidified environment at 37 °C. pcDNA3 plasmids of human a4 N-terminal domain (amino acid 1–421) wildtype and mutant K234A/K237A, 5 μg of plasmid/dish, were transfected into HEK293 cells using PolyJet Reagent (SignaGen, Frederick, MD, USA) in accordance with the procedure recommended by the manufacturer. Cells were harvested 30 h post transfection and pelleted by centrifugation (100× *g*). Cells were resuspended in homogenized buffer (250 mM sucrose, 1 mM EDTA, 10 mM HEPES, protease inhibitor cocktails) and lysed in a Dounce homogenizer. Lysates were separated by low-speed centrifugation (1000× *g*) for 10 min. Supernatants were subjected to ultracentrifugation (80,000× *g*) for 1 h to collect microsomal fractions and cytosolic fractions. Microsomal pellets were resuspended in 100 μL of homogenized buffer. The fractions were analyzed by immunoblots as described above.

### 4.7. Immunofluorescence

HEK293 cells were cultured as described above and seeded on coverslips in 6-well plates. Cells were co-transfected with plasmids of human a4 N-terminal domain (WT and mutants) and fluorescent proteins using PolyJet Reagent (SignaGen) in accordance with the procedure recommended by the manufacturer. Cells were fixed with 4% paraformaldehyde (PFA), permeabilized with 0.2% Triton-X and stained with anti-Flag antibody (Abcam, Cambridge, UK). For ionomycin treatment, cells were treated with 5 µM of ionomycin at room temperature for 15 min right before fixing. Images were acquired with a confocal microscope (Leica Confocal SP8, Wetzlar, Germany) using the 63× oil objective. Colocalization were measured with Mander’s coefficient M1, which calculates the percentage of total signal from the green channel which overlaps with the signal from the red channel [53].

### 4.8. Statistical Analysis

GraphPad Prism 9.4.1 software was used for statistical analysis and statistical graph production. One-way ANOVA followed by Dunnett’s multiple comparison test or Student’s *t*-test were used as indicated in figure legends. In figure, asterisks were used as follows: * indicates *p* < 0.05, ** indicates *p* < 0.01, *** indicates *p* < 0.001. The experimental results are expressed as the mean ± SEM.

## Figures and Tables

**Figure 1 ijms-24-04867-f001:**
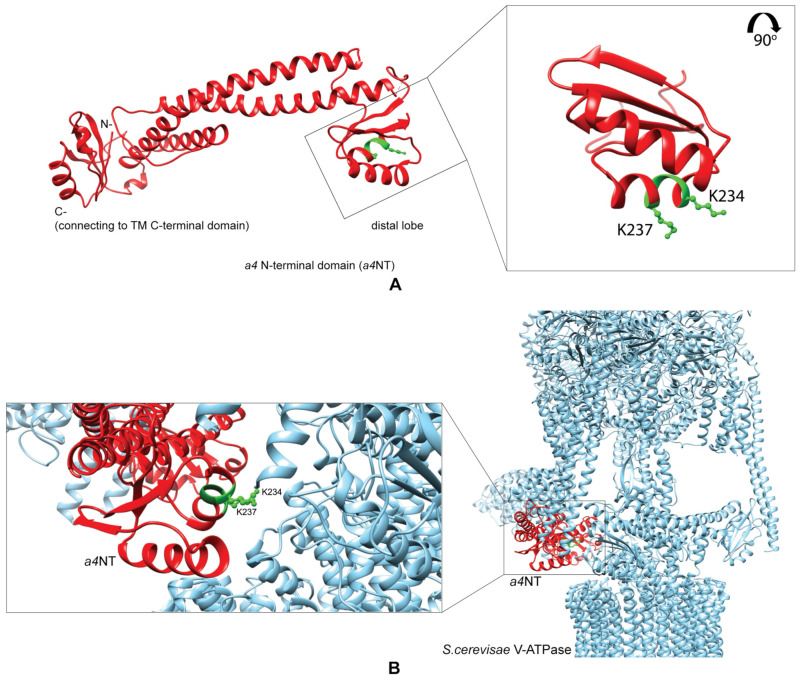
Homology model of a4NT reveals a putative lipid binding motif K^234^IKK^237^. (**A**) Homology model of the N-terminus of the a4 isoform was constructed using Phyre2.0. 32% i.d with *S. cerevisiae* V-ATPase state 1 Vph1p-NT (PDB: 3j9t.b). Inset: the putative phosphoinositide binding domain consists of three parallel β-strands and two α-helices. Highlighted in green is the conserved binding motif K^234^IKK^237^. (**B**) Fitting of the a4NT model (red) in the place of Vph1 N-terminal domain in the available cryo-EM generated structure of *S. cerevisae* V-ATPase (PDB: 3j9t) (cyan), with the PIP binding motif highlighted in green, and the side chain of two residues, K234 and K237, is shown to implicate the plasma membrane access (inset: enlarged view) (red: a4NT, cyan: *S. cerevisae* V-ATPase, green: key residues).

**Figure 2 ijms-24-04867-f002:**
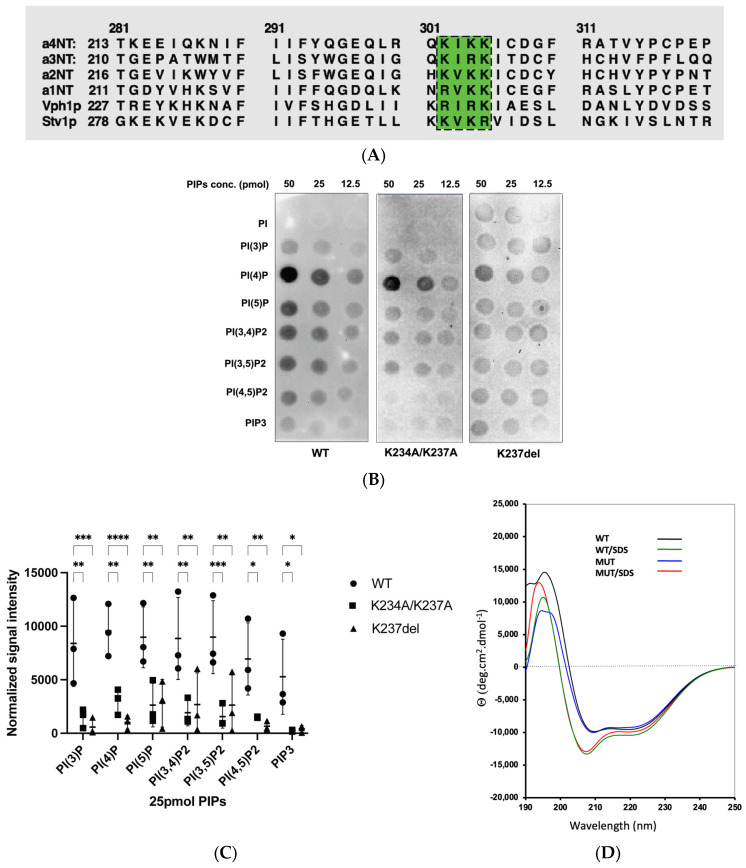
(**A**) PIP-binding motif (K/R)X(K/R)K is found within all four *a* isoforms. Sequence alignment of the N-terminal domains of the yeast Vph1p, Stv1p and human a1–a4. Highlighted in green is the binding motif (K/R)X(K/R)K. (**B**) Double mutation K234A/K237A and human dRTA mutation K237del show reduced PIP binding in protein–lipid overlay assay. Protein–phosphoinositide overlay assay of the cytoplasmic N-terminal domain of a4 WT and mutants (12 μg of proteins) with a PIPs array containing the dilution series of the indicated phosphoinositides (50, 25 and 12.5 pmol). (**C**) Quantification of overlay assay. Data represent mean ± S.D. of the signal intensity for 25 pmol of PIPs (normalized to total bound protein of the blots) from three independent experiments. Statistical significance was analyzed by one-way ANOVA with Dunnett’s multiple comparisons test. * Indicates *p* < 0.05, ** indicates *p* < 0.01, *** indicates *p* < 0.001, **** indicates *p* < 0.0001. (**D**) Mutations do not affect protein secondary structure. Circular dichroism (CD) spectra of a4NT wildtype (black) and mutant K234A/K237A (blue) in 50 mM Tris pH 8.0, and in addition of 10 mM SDS (wildtype: green; mutant: red).

**Figure 3 ijms-24-04867-f003:**
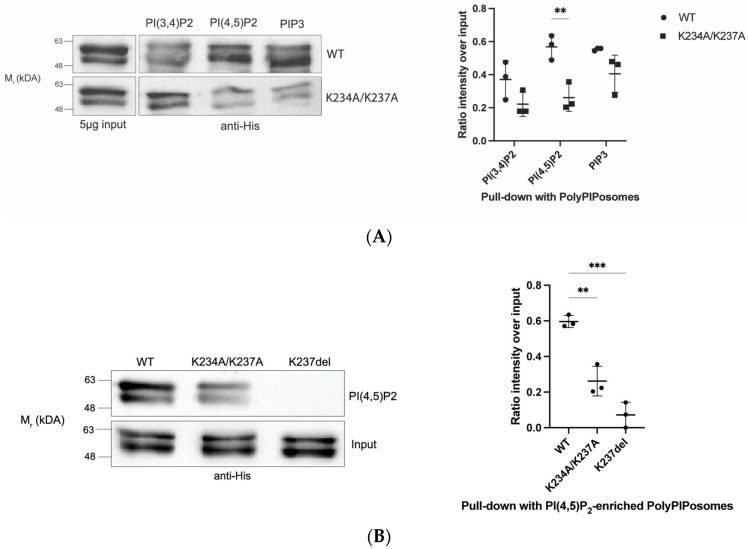
Mutation within the putative a4 PIP binding motif reduced interactions with PI(4,5)P_2_-enriched liposomes. (**A**) Liposome pull-down assay with PolyPIPosomes (Echelon) enriched with PI(3,4)P_2_, PI(4,5)P_2_ and PI(3,4,5)P_3_ of the WT and double mutant K234A/K237A. In total, 30 μg proteins (wildtype and mutant) was incubated for 1 h at room temperature with 20 μL of 1 mM PolyPIPosomes containing 5% of the PIP in binding buffer (50 mM Tris, pH 7.5, 150 mM NaCl, 0.05% Nonidet P-40). A total 5 μg of purified proteins was input as loading control. Quantification: Relative signal intensity of the proteins (WT, double mutants) pulled down with indicated liposomes to input. Data represent mean ± S.D. from three independent experiments. Pair *t*-tests were run to analyze the significance in mean difference. ** indicates *p* < 0.01. (**B**) Liposome pull-down assay with PI(4,5)P_2_-enriched PolyPIPosome with WT and K234A/K237A and K237del mutants. Quantification: relative intensity of the proteins pulled down with PI(4,5)P2-enriched liposome to input. Data represent mean ± S.D. from three independent experiments. Statistical significance was analyzed by one-way ANOVA with Dunnett’s multiple comparisons test comparing mutants to WT. ** indicates *p* < 0.01, *** indicates *p* < 0.001. (**C**) Liposome pull-down assay with PI(4)P-enriched PolyPIPosome with WT and K237del mutant. Quantification: relative intensity of the proteins pulled down with the liposome to input (30 ng). Data represent mean ± S.D. from five independent experiments.

**Figure 4 ijms-24-04867-f004:**
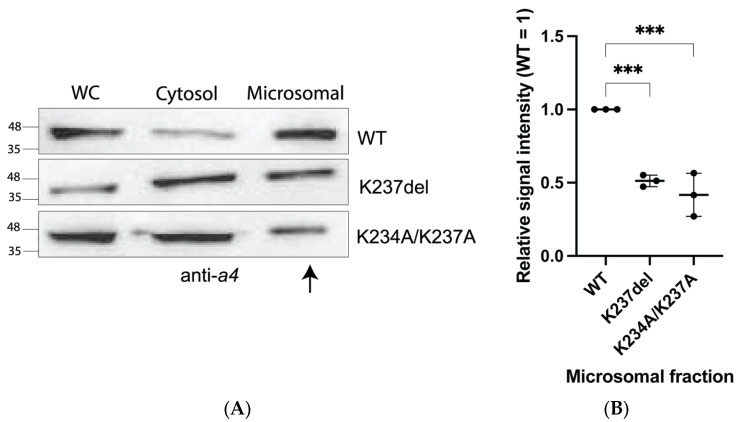
Mutation K234A/K237A and K237del reduced a4NT in microsomal fractions. (**A**) a4NT-WT co-purified with microsomal membrane while mutants a4NT-K234A/K237A and K237del reduced membrane association. Plasmids containing *a4*NT wildtype (WT) and mutants K234A/K237A and K237del were transfected in HEK293 cells. Cells were harvested after 30 h. Whole-cell extracts (WC) were centrifuged at 48,000 rpm to obtain cytosolic fraction (cytosol) and microsomal fraction (microsomal). (**B**) Quantification. Relative pixel intensity of microsomal fraction (WT microsomal = 1). Data represent mean ± S.D. from three independent experiments. Statistical significance was analyzed by one-way ANOVA with Dunnett’s multiple comparisons test comparing mutants to WT. *** indicates *p* < 0.001.

**Figure 5 ijms-24-04867-f005:**
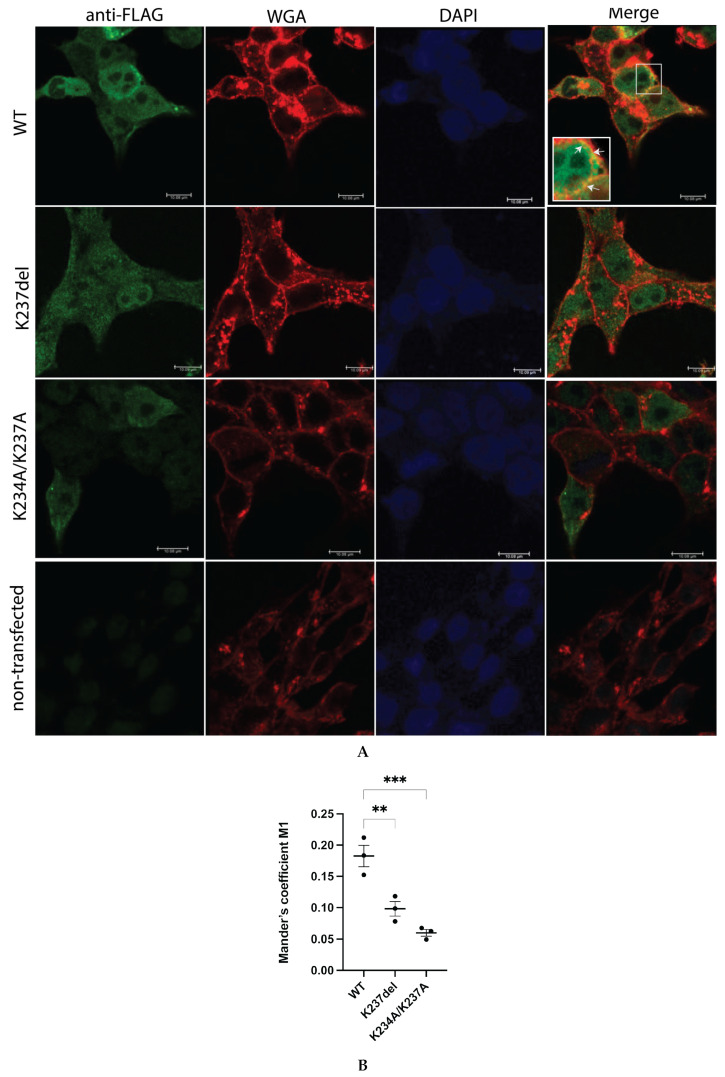
Mutation K234A/K237A and K237del reduced plasma-membrane-associated a4NT. (**A**) Plasmids containing FLAG-tagged a4NT wildtype (WT), mutants K234A/K237A and K237del were transfected in HEK293 cells. Cells were fixed and stained for membrane marker Alexa-647 conjugated wheat germ agglutinin (WGA) (red), followed by permeabilization and staining for FLAG-tagged proteins (green). White arrows indicate regions of plasma membrane localization. The scale bar represents 10 μm (inset at the top panel is enlarged area of colocalization). (**B**) Quantification of the green signal intensity at the vicinity of the plasma membrane (red). Data represent mean value of at least 20 cells assessed ± SEM from three independent experiments. Statistical significance was analyzed by one-way ANOVA with Dunnett’s multiple comparisons test comparing mutants to WT. ** indicates *p* < 0.01, *** indicates *p* < 0.001.

**Figure 6 ijms-24-04867-f006:**
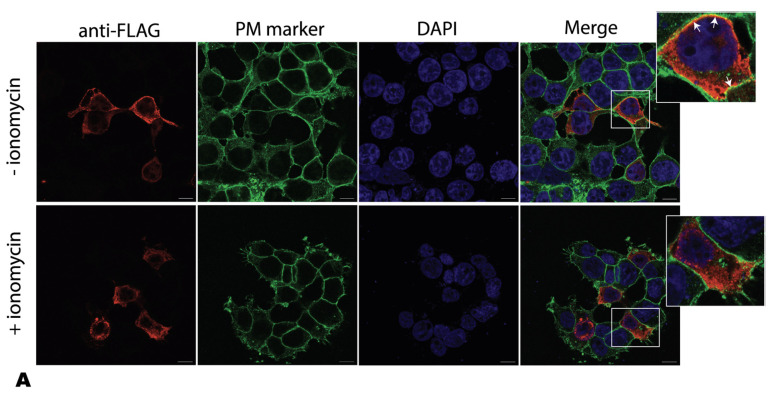
Ionomycin reduces a4NT-membrane association in vivo. (**A**) Plasmids containing FLAG-tagged a4NT wildtype (WT) were transfected in HEK293 cells. Cells were treated with 5 µM of ionomycin at room temperature for 15 min before harvest. Cells were fixed and permeabilized, then stained for membrane marker anti-Na^+^/K^+^-ATPase with Alexa Fluor 488 (green), followed by staining for FLAG-tagged proteins (red). White arrows represent regions of plasma membrane localization. Images are representatives of 20 cells each from 4 experiments. The scale bar represents 10 μm (inset at the right panel is enlarged area of colocalization). (**B**) Quantification. Relative intensity of the red signal overlapping with the green signal to the total red signal. Data represent mean value of at least 20 cells assessed ±SEM from 4 independent experiments. A pair *t*-test was run to analyze the significance in mean difference. ** indicates *p* < 0.001. (**C**) Treatment with ionomycin causes depletion of membrane PI(4,5)P_2_, resulting in mislocalization of PH-PLC. Plasmids containing GFP-tagged PH-PLC were transfected in HEK293 cells. Cells were treated with 5 µM of ionomycin at room temperature for 15 min before harvest. Cells were fixed and permeabilized, then stained for membrane marker anti-Na^+^/K^+^-ATPase with Alexa Fluor 568 (red). Quantification of the resulting immunofluorescent images were performed by measuring the relative intensity of the red signal overlapping with the green signal to the total red signal. Data represent mean value of at least 20 cells assessed ±SEM from 3 independent experiments. A pair *t*-test was run to analyze the significance in mean difference. * indicates *p* < 0.05.

## Data Availability

All data generated or analyzed during this study are included in this published article.

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
