# Peer review of "Characterization of a PIP Binding Site in the N-Terminal Domain of V-ATPase a4 and Its Role in Plasma Membrane Association"

_ijms, 2023, doi:10.3390/ijms24054867_

Round 1
Reviewer 1 Report
Chu et al reported the model structure and function of basic motif (K234IKK237) of V-ATPases as multi-subunit ATP–dependent proton pumps involved in various cellular functions.The flow of the manuscript is relatively scientific, but its function was in fact predictable, as the authors noted “critical for interaction with phosphoinositides (PIP) and find similar basic residue motifs in all four 22 mammalian and both yeast a subunit isoforms.” So I think it's low novelty. Accordingly, in order to make up for this shortcoming, the author will have to write more discussion about the importance of the mutation site. On the other hand, the author's structural modeling and explanation using it are very basic and provide limited information.
1. The research result is not highly novel because it was predictable to some extent through previous reports. In order to make up for this, the author will have to do more discussion about the novelty of the research result of the author using the previous paper reported.
2. Line 38-44: “0. How to Use This Template The template details the sections that can be used in a manuscript. Note that each section has a corresponding style, which can be found in the “Styles” menu of Word. Sections that are not mandatory are listed as such. The section titles given are for articles. Review papers and other article types have a more flexible structure. Remove this paragraph and start section numbering with 1. For any questions, please contact the editorial office of the journal or support@mdpi.com. “
Author should delete this phrase.
3. Requires many modifications to Figure 1.
Figure 1a. Author should mark N-terminal and N-terminal in the protein structure.
Figure 1b. Since the authors used a model structure, comparisons with other proteins are meaningless. Also, there is no sufficient description of the Osh3 PH domain in the text. In addition, I do not think that the main chain comparison with the homology model is meaningful, and it will be necessary to compare major amino acid sites.
Figure 1c. The unnecessary outline of Figure 1c should be removed. Author should indicate which molecule it is.
On the other hand, how reliable is the model structure of Phyre2.0 used by the author? Since the model used and the amino acid sequence is 32% identical, the position of the main chain may be similar, but the detailed side chain of amino acid conformation is expected to be different. In addition, the author introduced the binding site by superimposing the a4.NT model on S.cereviase V-ATPase, but it seems necessary to explain the structure of S.cereviase V-ATPase using the a4.NT model at the same location.
On the other hand, I think that the author's modeling part will be more reliable if the author uses Alphafold, which has recently been in the limelight, rather than Phyre2.0, to distinguish the reliable interval (confidence values) for each amino acid in the model structure.
4. Line 147: “Alignment was generated by 147 Chimera.” This should go to method section.
5. Line 239: “E. coli” This should be italicized.
6. Line 353-359: “4.2. Protein structure prediction. The homology model of human a4NT (1-421) was obtained by submission of the sequence to the I-TASSER online threading programs, with constraint coordinate from the crystal structure of Meiothermus ruber A-ATPase subunit I (PDB: 3rrk). The structural alignment of the a4NT model and Osh3 lipid binding domain (PDB: 3vu4) was done using UCSF Chimera. Sequence alignment of all 4 human an isoform and yeast Vph1p, Stv1p were obtained by Chimera.” Authors should add references to programs and structures used. Also, the PDB codes mentioned in the result and method are different.
7. Line 420-421: Authors should remove underlines from funding information.
8. Reference: Authors should correct any missing volume or page information.
Author Response
1/ Review: The research result is not highly novel because it was predictable to some extent through previous reports. In order to make up for this, the author will have to do more discussion about the novelty of the research result of the author using the previous paper reported.
Response: Added to discussion "In a yeast study, Banerjee et al suggested that residue K84 within Stv1p is critical for PI(4)P interaction [22]. The K84 residue resides on a flexible loop exposed to the membrane and is within the Stv1p Golgi localization signal – W83KY [21], however this residue is not conserved in human and Vph1p. Studies of PIP binding domains indicate that highly conserve basic motifs are necessary, but not exclusive, for interaction with PIPs [51]. Further, protein folding play an additional key role in bringing distant residues in the sequence to closer proximity in 3D structure, which altogether strengthen the protein-lipid interactions. We hypothesize that in Stv1p, the putative binding motif and the K84 residues work cooperatively to strengthen and/or define the specificity of the protein and lipid interaction."
2/Line 38-44:“0. How to Use This Template The template details the sections that can be used in a manuscript. Note that each section has a corresponding style, which can be found in the “Styles” menu of Word. Sections that are not mandatory are listed as such. The section titles given are for articles. Review papers and other article types have a more flexible structure. Remove this paragraph and start section numbering with 1. For any questions, please contact the editorial office of the journal or support@mdpi.com. “
Author should delete this phrase.
Response: Removed the template instruction.
3/ Review: Modifications to Figure 1.
Response: Remove Fig 1B as recommended. See revised version for the modified Fig 1.
Figure 1: Homology model of a4NT reveals a putative lipid binding motif K234IKK237 . (A) Homology model of the N-terminus of the a4 isoform was constructed using Phyre2.0. 32% i.d with S.cerevisiae V-ATPase state 1 Vph1p-NT (PDB: 3j9t.b). Inset: the putative phosphoinositide binding domain consists of 3 parallel β-strands, and 2 α-helices. Highlighted in green is the conserved binding motif K234IKK237. (B) Fitting of the a4.NT model (red) in the place of Vph1 N-terminal domain in the available cryo-EM generated structure of S.cerevisae V-ATPase (PDB: 3j9t) (cyan), with the PIP binding motif highlighted in green, and the side chain of two residues K234, and K237 was shown to implicate the plasma membrane access (inset: enlarged view).
4/ Line 147: “Alignment was generated by 147 Chimera.” This should go to method section.
Response: Moved to Method section "Sequence alignment of all 4 human a isoform and yeast Vph1p, Stv1p were obtained by Chimera".
5/Line 239: “E. coli” This should be italicized.
Response: Change made in the revised version.
6/ Line 353-359: “4.2. Protein structure prediction. The homology model of human a4NT (1-421) was obtained by submission of the sequence to the I-TASSER online threading programs, with constraint coordinate from the crystal structure of Meiothermus ruber A-ATPase subunit I (PDB: 3rrk). The structural alignment of the a4NT model and Osh3 lipid binding domain (PDB: 3vu4) was done using UCSF Chimera. Sequence alignment of all 4 human an isoform and yeast Vph1p, Stv1p were obtained by Chimera.” Authors should add references to programs and structures used. Also, the PDB codes mentioned in the result and method are different.
Response: Edited section 4.3.
Homology model of the N-terminus of the a4 isoform (amino acid 1-420) was constructed using Phyre2.0 with reference coordinates of S.cerevisiae V-ATPase state 1 Vph1p-NT (PDB: 3j9t.b). Confidence interval of the model is 100%. Sequence alignment of all 4 human a isoform and yeast Vph1p, Stv1p were obtained by Chimera.
7/Line 420-421: Authors should remove underlines from funding information.
Response: Change made in text in revised version
8/Reference: Authors should correct any missing volume or page information.
Response: Change made in text in revised version
Reviewer 2 Report
V-ATPases are multisubunit, ATP-dependent proton pumps found in intracellular and plasma membranes of specialized cells. In recent years, there has been an increase in evidence implicating the V-ATPase in cancer cell invasiveness.
Understanding their association with membrane is of special interest.
Here the authors have nicely identified and characterized a region responsible for membrane association of V-ATPase a4.
The work is well conducted and presented as well as convincing.
Delete paragraph "how to use this template" before introduction.
Author Response
Reviewer: Delete paragraph "how to use this template" before introduction.
Response: Deleted!
Reviewer 3 Report
Dear Authors,
Reviewer comments IJMS-2217656
The manuscript enttiled „Characterization of a PIP binding site in the N-terminal domain of V-ATPase a4 and its role in plasma membrane association“ represents a valuable contribution to an analysis of molecular characteristics of human V-ATPase a4 isoform with respect to its plasma membrane binding properties and its interactions with PIP as membrane lipid signalling molecule phosphatidylinositol.
The manuscript provides a nice molecular analysis of VTPase a4 isoform aimed at determination of the protein structure, protein-phospholipid interaction studied by using protein-lipid overlay assays and subcellular localization using immunofluorescence.
I can therefore recommend the manuscript for publication in International Journal of Molecular Sciences.
However, I have a few important comments on the present mansucript version which are gievn below:
1/ Terminology: Use the term „immunoblotting“ instead of „Western blotting“.
Units: Write „1 h“ (not „1 hr“), „2 min“ (not „2 mins“).
2/ Materials and methods:
In Materials and methods section, the sources of all experimental materials used, i.e., E. coli Rosetta culture (line 340), HEK293 cells (line 384) as well as human a4NT protein wild-type and its mutant forms have to be given.
3/ Statistical analysis of the data:
At the end of Materials and methods section, a short section on Statistical analysis of the presented data has to be added as part “4.7. Statistical analysis of the data“ including the approaches used (such as paired T-test, ANOVA followedby multiple-comparisons test) and software used. The kind of multiple-comparisons test used for the determination of significant differences following ANOVA analysis has to be given in Materials and methods as well as in Results sections.
4/ Ethics statement: Since the authors worked with human protein I think that they need to add some Ethics statement in the manuscript. I personally work with plant proteins only so I do not need any ethics statement but I think that the researchers working with animal and human materials need them.
5/ Formal comments on the text:
Results, line 129: In 2.2. heading, modify the statement „with PI(4,5)P2-enriched liposomes“ (not „with PI(4,5)P2-enrich liposomes“).
Results, line 169: Write either „there was no significant difference in the binding between wildtype and K237del (Fig. 3C).“ or „there were no significant differences in the binding between wildtype and K237del (Fig. 3C).“
Results, line 177: In Figure 3 legend, modify the statement „with PI(4,5)P2-enriched liposomes“ (not „with PI(4,5)P2-enrich liposomes“).
Results, in Figure 3, Figure 4 and Figure 5 legends, the kind of multiple comparisons test following ANOVA analysis has to be given.
Results, lines 193, 194: Add a space between the number and the corresponding unit in „222 nm and 208 nm“.
Results, line 232: Add the word „fraction“ following the word „the cytosolic“ in the statement „…both mutant proteins were mainly in the cytosolic fraction and did not appear in the vicinity of the plasma membrane…..“
Results, Figure 6 legend, line 280: Remove the extra „with“ in the statement „Cells were treated with 5 µM of ionomycin at room temperature…“
Discussion, line 287: Add the full protein name in the statement „Here, we show that the Na+/K+-ATPase a4 isoform,….“
Discussion, line 319: Correct the typing error in the word „within“ (not „withing“).
Discussion, line 329: Use the word „enhanced“ instead of „higher“ in the statement „enhanced a4 and a4 expression and upregulation of PI(4,5)P2 were both critical for formation of invadopodia and cell invasion…“
Materials and methods, line 343: Correct the word form „Cells“ (not „Celled“) in the statement „Cells were harvested by centrifugation,….“
Materials and methods, lines 347-350: Add a space between the number and the corresponding unit in „5 µg/mL“, „20 mM“, „50 mM“, etc.
Materials and methods, line 358: Correct the word forms in the statement „Sequence alignment of all 4 human a isoforms…“ (not „all 4 human isoform…“).
Materials and methods, lines 366-369: Write „h“ instead of „hr“, „min“ instead of „mins“, and add a space between a number and a corresponding unit in „30 s.“
Materials and methods, lines 382, 395: Use the term „immunoblotting“ instead of „Western blotting“.
Materials and methods, line 391: Correct the word form in the term „in homogenization buffer“ (not „homogenize buffer“).
Materials and methods, line 395: Modify teh word form „describe“ to „described“ in the statement „The fractions were analysed by immunoblot as described above.“
Final recommendation: Accept after a minor revision.
Author Response
1/ Terminology:
Review: Terminology: Use the term „immunoblotting“ instead of „Western blotting“.
Response: Replaced "Western blotting" with "immunoblotting"
Review: Units: Write „1 h“ (not „1 hr“), „2 min“ (not „2 mins“).
Response: All units were edited.
2/ Materials and methods:
Review: In Materials and methods section, the sources of all experimental materials used, i.e., E. coli Rosetta culture (line 340), HEK293 cells (line 384) as well as human a4NT protein wild-type and its mutant forms have to be given.
Response: Add the vendors for all material used. E.coli Rosetta (DE3) (Sigma-Aldrich); HEK293 cells (ATCC).
Adding section 4.1. Plasmids: pET32a+ :: human V-ATPase a4NT (MM1111). The PCR products of N-terminal domain of human a4 with primers MO503: 5’-ACGTGGTACCATGGTGTCTGTGTTTCGAAG and MO504: 3’-ACGTGAATTCCTCCGGTGTAGGGGGCTGGGTTTATC, from start codon to T397 was cloned into pcDNA3 and pET32a(+) plasmids between KpnI and EcoRI sites, respectively, resulting MM1071 and MM1103; and PCR products of human a4 with primers MO521: 5’-GCAACATCGACGTCACCCAGCAG-3’ and MO522: 5’-ACGTGAATTCTTAGGTGTAGGGGGCTGGGTTTATC-3’ was used to replace the region from ZraI to EcoRI (at position V316 to the 3’ end) in MM1103, which adding a stop codon TAA just after T397, resulting MM1111. pET32a+ :: human V-ATPase a4NT K234A/K237A (MM1113) A designed a4NT K234A/K237A double mutation was synthesized by GeneArt Gene Synthesis, Invitrogen Canada and delivered in the form of pMA-T::partial a4NT DNA fragment covered from BamHI to EcoNI sites of human a4NT. The synthesized DNA fragment was used to replace the same region of MM 1071, resulting MM1097; when the whole KpnI and EcoRI insert of a4NT in MM1097 was moved into pET32a(+), the new construct was named as MM1105. A TAA stop codon after T397 was added in MM1105 and resulting the final MM1113. pET32a+ :: human V-ATPase a4NT ∆K237 (MM1130) Q5 Site-Directed Mutagenesis Kit (NEB E0554) was used on MM1111 to make human V-ATPase a4NT.∆K237 mutant with primers MO527: 5’-ATCTGTGATGGGTTTCGAG and MO528: 3’-CTTGATTTTCTGCCTGAG, sequencing verified. pcDNA3 :: human V-ATPase a4NT (MM1123) KpnI and EcoRI insert of a4NT in MM1111 was moved into pcDNA3.1+, the new construct was named as MM1123. pcDNA3 :: human V-ATPase a4NT K234A/K237A (MM1125) KpnI and EcoRI insert of MM1113 was ligated into pcDNA3.1+, the new construct was named as MM1125. pcDNA3 :: human V-ATPase a4NT ∆K237 (MM1131) KpnI and EcoRI insert of MM1130 was ligated into pcDNA3.1+, the new construct was named as MM1131.
3/ Statistical analysis of the data:
Review: At the end of Materials and methods section, a short section on Statistical analysis of the presented data has to be added as part “4.7. Statistical analysis of the data“ including the approaches used (such as paired T-test, ANOVA followed by multiple-comparisons test) and software used. The kind of multiple-comparisons test used for the determination of significant differences following ANOVA analysis has to be given in Materials and methods as well as in Results sections.
Response: Added section 4.8. Statistical Analysis: GraphPad Prism 9.4.1 software was used for statistical analysis and statistical graph production. One-way ANOVA followed by Dunnett’s multiple comparison test or Student’s t-test were used as indicated in figure legends. In figure, asterisks were used as follows: * indicates p<0.05, ** indicates p<0.01, *** indicates p<0.001. The experimental results are expressed as the mean ± SEM.
4/ Ethics statement:
Review: Since the authors worked with human protein I think that they need to add some Ethics statement in the manuscript. I personally work with plant proteins only so I do not need any ethics statement but I think that the researchers working with animal and human materials need them.
Response: Added Ethical Statements: The HEK293 cell line was obtained from ATCC. Human a4NT plasmid constructs were cloned in the lab as described above.
5/ Formal comments on the text:
Response: All changes were made accordingly in revised version.
Round 2
Reviewer 1 Report
The authors addressed all of the reviewer's concerns. The revised manuscript is suitable for publication.